# Regulation of RAD51 at the Transcriptional and Functional Levels: What Prospects for Cancer Therapy?

**DOI:** 10.3390/cancers13122930

**Published:** 2021-06-11

**Authors:** Esin Orhan, Carolina Velazquez, Imene Tabet, Claude Sardet, Charles Theillet

**Affiliations:** 1IRCM, Institut de Recherche en Cancérologie de Montpellier U1194 INSERM, Université de Montpellier, 34090 Montpellier, France; esin.orhan@inserm.fr (E.O.); imene.taber@inserm.fr (I.T.); claude.sardet@inserm.fr (C.S.); 2ICM, Institut du Cancer de Montpellier, 34090 Montpellier, France; carolina.velazquez@inserm.fr

**Keywords:** RAD51, homologous recombination, DNA repair, genomic instability, DNA break, cancer therapy

## Abstract

**Simple Summary:**

RAD51 is an essential gene for cell survival. Its function is central to DNA repair and it protects cells from life-threatening damage to the genome. Interestingly, RAD51 is expressed at high levels in a large proportion of cancers, and elevated RAD51 expression is associated with a bad outcome and reduced response to treatment. Hence, reducing RAD51 expression and/or interfering with its function could be of great therapeutic value. We review here the multiple levels of regulation of RAD51 expression and function and explore potential therapeutic leads.

**Abstract:**

The RAD51 recombinase is a critical effector of Homologous Recombination (HR), which is an essential DNA repair mechanism for double-strand breaks. The RAD51 protein is recruited onto the DNA break by BRCA2 and forms homopolymeric filaments that invade the homologous chromatid and use it as a template for repair. RAD51 filaments are detectable by immunofluorescence as distinct foci in the cell nucleus, and their presence is a read out of HR proficiency. RAD51 is an essential gene, protecting cells from genetic instability. Its expression is low and tightly regulated in normal cells and, contrastingly, elevated in a large fraction of cancers, where its level of expression and activity have been linked with sensitivity to genotoxic treatment. In particular, BRCA-deficient tumors show reduced or obliterated RAD51 foci formation and increased sensitivity to platinum salt or PARP inhibitors. However, resistance to treatment sets in rapidly and is frequently based on a complete or partial restoration of RAD51 foci formation. Consequently, RAD51 could be a highly valuable therapeutic target. Here, we review the multiple levels of regulation that impact the transcription of the RAD51 gene, as well as the post-translational modifications that determine its expression level, recruitment on DNA damage sites and the efficient formation of homofilaments. Some of these regulation levels may be targeted and their impact on cancer cell survival discussed.

## 1. Introduction

DNA damage inflicted by exogenous or endogenous factors such as oxidized radicals, ultra violet (UV) light, ionizing radiation (IR) or genotoxic chemicals are a constant threat to genome integrity. Complex and efficient DNA repair mechanisms preventing the transmission of genetic errors to the progeny are already present in bacteria and have evolved in complexity and specificity in higher eukaryotes. Deficiency in one of the multiple repair mechanisms inevitably results in the accumulation of mutations. DNA repair mechanisms are highly specialized according to the type of damage inflicted. In case of DNA breaks, mechanisms depend on the nature of the break (single or double-strand break), as well as on the cell cycle phase. Unrepaired double-strand breaks (DSB) are highly toxic because they open the door to illegitimate chromosomal rearrangements and can lead to cancer [1]. To keep DNA integrity and stability, cells need to accurately repair these lesions.

Principal DSB repair mechanisms are non-homologous end joining (NHEJ) repair and homologous recombination (HR) [2]. NHEJ is based on minimal homology of the sequences around the break site in order to ligate the DNA ends. NHEJ is considered an error-prone mechanism that is active throughout the whole cell cycle. During S and G2 phases, HR is activated and competes with NHEJ [3]. Because HR uses the sister chromatid as a template, it is more accurate and avoids misalignment and illegitimate rearrangements that are a risk with NHEJ [4]. Furthermore, DSBs are more susceptible to occurring during the S phase, because of replication fork collapse. For these reasons, HR repair is essential for the integrity of the genome and cell survival. In case of HR deficiency, cells become vulnerable to DSB-induced genome instability [5,6].

When a DSB occurs, the DNA-damage sensors Ataxia-Telangiectasia-Mutated (ATM) and Ataxia Telangiectasia and Rad3-related (ATR) kinases phosphorylate Histone H2AX [7,8]. After this initial signal, DNA repair proteins are recruited to the break site. During the G1 phase of the cell cycle, TP53-Binding Protein (TP53BP1, also called 53BP1) and the shielding complex are recruited at the break site and allow for the recruitment of NHEJ actors, such as DNA-dependent Protein Kinase catalytic subunit (DNA-PKcs), KU70/80, XRCC4 and LIG4 [9,10]. During the S/G2 phases of the cell cycle, Breast Cancer 1 (BRCA1) is phosphorylated and activated by the Cyclin Dependent Kinase 1 (CDK1) [11] and, in the presence of DSB, BRCA1 is also phosphorylated by Checkpoint Kinase 2 (CHK2) that signals the start of the HR repair (HRR) cascade [12]. BRCA1 binds to CtBP-interacting protein (CtIP) that competitively removes 53BP1 from the DSB site and engages HRR [13,14]. CtIP recruits the MRN exonuclease complex which resects the DNA from 5′ to 3′, leaving long 3′ single-strand DNA (ssDNA) overhangs on either side of the DSB [15]. The ssDNA overhangs are rapidly coated and protected by Replication Protein A (RPA). BRCA1 recruits BRCA2 at the break site together with the bridge protein Partner and Localizer of BRCA2 (PALB2). Then, BRCA2 recruits the RAD51 recombinase, which is a crucial protein for the HR to occur. The RAD51 protein forms homopolymeric filaments around the ssDNA and removes RPA. Then, it proceeds to the invasion of the sister chromatid and searches for the matching homologous sequence (Figure 1) [16].

The RAD51 protein is, thus, key for the DNA-strand exchange steps, which HRR is dependent on. RAD51 belongs to the RAD51 protein family that is characterized by a ~230 amino acid domain containing two ATPase motifs called Walker A and Walker B [17]. Human RAD51 actively binds to ssDNA via its N-terminal portion and mediates HR between homologous DNA strands (Figure 2) [18]. RAD51 can be activated by several post-translational modifications, which control its activity at different steps of HRR [19]. We will describe these processes in detail.

RAD51 also plays a role in the repair and re-start of the stalled DNA replication forks, which, in the absence of repair, collapse and leave long stretches of unprotected ssDNA leading to DSB. One of the mechanisms to protect stalled replication forks is replication fork reversal, in which RAD51 is critical [20]. While in normal cells fork reversal is finely tuned to avoid excessive recombination, cancer cells treated with genotoxic agents are challenged with a high number of stalled forks that can overwhelm DNA repair. Thus, elevated levels of RAD51 expression could act as a safeguard to avoid massive fork collapse and protect cancer cells from the accumulation of DNA breaks.

In normal human cells, the expression and activity of RAD51 are tightly controlled in order to avoid aberrant DNA recombination [21]. It is of note that RAD51 expression is strongly increased in breast, thyroid or pancreatic cancer [22,23,24] and that its overexpression is strongly correlated with poor prognosis [25,26]. Over-activated HRR has been proposed to increase the genetic plasticity of cancer cells due to elevated recombination [21,27]. It is also thought to increase the capacity to repair DNA breaks and, hence, protect cells from DNA-damaging treatments. Indeed, elevated RAD51 expression has been correlated with resistance to chemotherapy [28,29]. Conversely, BRCA-deficient ovarian cancers, which are HR-deficient, show increased sensitivity to platinum salts and Poly (ADP-ribose) polymerase inhibitors (PARPi) [30]. However, resistance to treatment occurs quickly and, in the case of PARPi, arises mainly by restoring HRR despite BRCA-deficiency. In some studies, restored HRR is monitored by the presence of RAD51 foci in nuclei after treatment-induced DNA damage [31]. This led some authors to propose the use of RAD51 expression or foci detection as a predicting biomarker of HR proficiency to orient the choice of treatments.

The need of alternative treatments for cancer patients that do not respond or develop resistance to DNA-damage-inducing treatments has become evident. RAD51 downregulation has been proposed as a strategy to overcome drug resistance [29]. Thus, better apprehension of RAD51 activity and regulation could lead to new therapeutic strategies for resistant tumors. Several inhibitors of potential RAD51 regulators, either alone or in combination with DNA-damaging agents, have been tested in clinical trials listed in the Appendix A (Appendix A). In this review, we highlight the importance of RAD51 regulators at the transcriptional and post-translational levels and potential implications for novel treatment strategies. We discuss the clinical significance of RAD51 as a biomarker for HR capacity.

## 2. Regulation of RAD51 Expression

Transcription factors that regulate RAD51 transcription could, thus, be key in the response of cancer cells to DNA-damaging treatments (Figure 3).

### 2.1. Transcriptional Control of RAD51

#### 2.1.1. Cyclin Dependent Kinases

Cyclin-dependent kinases (CDK) have been proposed to play a role in the transcriptional regulation of RAD51. CDKs are a large family of genes primarily known to regulate cell transition from a cell cycle phase to the next. However, CDK7-13 have been shown to act as key regulators of transcription by way of phosphorylation of the C-terminal domain (CTD) of RNA polymerase II at different serine residues and promote the recruitment of factors necessary for full initiation and elongation of transcripts [32]. During gene transcription by RNA pol II, CDK7, a subunit of TFIIH, phosphorylates Ser5 and activates transcription initiation, whereas CDK9, which forms the pTEF complex with Cyclin T, phosphorylates Ser2 and catalyzes the elongation of the transcripts [33,34]. Interestingly, genetic knockdown or pharmacological inhibition of CDK9 revealed that highly transcribed genes associated with super-enhancers, such as oncogenes or anti-apoptotic genes, were preferentially impacted in cancer cells [34].

Another actively studied CTD Ser2 kinase, whose role was initially discovered in Drosophila, is CDK12 [35]. CDK12, which associates with cyclin K (CCNK), was also shown to regulate transcription elongation. Remarkably, the genetic depletion or pharmacological inhibition of CDK12 appeared to mainly affect long genes comprising multiple exons. This particularity has been attributed to the fact that CDK12 suppresses intronic polyadenylation, thus allowing for the full-length transcription of these long genes [36,37,38]. CDK13, which has strong sequence homology with CDK12, has been predicted to share a similar role in pre-mRNA processing [36].

CDK12 and its homolog, CDK13, have been recently proposed to regulate specifically the expression of DNA damage response genes (Table 1) [36,39,40]. Indeed, inhibition of CDK12 either by a drug or by small interfering RNAs (siRNAs) was shown to selectively decrease the expression of DNA damage response genes, such as BRCA1, ATM, FANCD2 and FANCI, possibly because these are large genes with multiple exons [36,39]. Interestingly, Querada and coworkers (2019) suggested that RAD51 expression could be regulated by CDK13, while BRCA1, FANCD2 and ATM transcription was dependent on CDK12 [40]. It thus seems that the knockdown of CDK12/13 affects the transcription of a smaller set of genes compared with the knockdown of CDK9, and it is not yet entirely clear how much both lists overlap. However, the increasing evidence of CDK12 mutations in prostate, triple-negative breast and more so in ovarian cancer has put the focus on this gene that is not considered to belong to the HR pathway per se [41,42,43]. Indeed, in The Cancer Genome Atlas (TCGA) dataset, 3% of high-grade serous ovarian cancer harbored a CDK12 mutation, an incidence similar to that observed for BRCA1 and BRCA2 (3.5 and 3.2%, respectively) [44]. Furthermore, consistent with a role of CDK12 in the regulation of HR genes, a combination of mechanistic studies with clinical evidence indicated that CDK12-driven transcription of HR genes is crucial for genome stability [45,46,47,48]. Hence, CDK12 could be a target of choice to exploit cancer vulnerability associated with DNA damage. Although further understanding of the impact and specificity of CDK inhibition on HR gene expression and activity is needed, it can be hypothesized that the use of selective inhibitors of these kinases could be a strategy of choice to sensitize cancer cells to genotoxic treatment [47]. This is supported by a number of studies on cell lines and preclinical models, showing that PARP-inhibitor (PARP-i) administration combined with a CDK12/13 inhibitor (CDK12-i), such as Dinaciclib or SR-4835, is synergistic compared with PARP-inhibitor alone [40,48,49,50]. Furthermore, the PARP-i/CDK12-i combination has been suggested to reverse acquired PARP-i resistance [48]. An ongoing clinical trial involving 63 triple-negative breast patients treated with Dinaciclib combined with Velaparib will give us the first clues on the benefits (efficacy and tolerability) of this regimen (Appendix A) [50].

#### 2.1.2. Member of the E2F Transcription Factor Family

Other documented transcription regulators of RAD51 are E2F family members, which are central gatekeepers of cell cycle progression and survival. Their transcriptional activity is regulated by the binding of retinoblastoma (RB) protein and pocket proteins, p107 and p130 [51,52]. Noticeably, E2F1, E2F4 or E2F7 are activated and stabilized upon DNA damage, possibly by way of phosphorylation by ATM, ATR and CHK2 [53,54]. Elevated E2F1 expression levels have been associated with poor prognosis in non-small cell lung carcinomas and gastrointestinal carcinomas [55,56]. In colon cancer cell models, knockdown of E2F1 was correlated with loss of RAD51 expression and RAD51-dependent DSB repair [57]; furthermore, E2F1 and BRIT1 (MCPH1) were shown to form a complex that activates the transcription of BRCA1, CHK1, p73 and CASP7 [58]. These data strongly suggest that E2F1 could have a role in both activating the transcription of HR genes and their recruitment to DSB.

E2F4 could also play a role in the transcriptional regulation of RAD51. Under hypoxic conditions, E2F4 binds to p130 and represses BRCA1 and RAD51 transcription [59]. The E2F4/p130 driven repression of HR genes could be one of the mechanisms explaining genetic instability in hypoxic cells, a liability that could be exploited to design cancer therapy strategies.

E2F7 has been shown to orchestrate transcription under chemotherapy-induced DNA damage. In such conditions, it binds to the promoter of RAD51, BARD1, CtIP, BRIP1 and FANCI and represses their expression. Depletion of E2F7 in U2OS human osteosarcoma cells resulted in increased expression of DNA repair pathway genes, enhanced HRR and replication fork stability [60]. In BRCA2-deficient cells, loss of E2F7 favored resistance to the PARPi Olaparib [61].

Finally, we must mention that CDK2 and CDK4/6 are direct regulators of the transcriptional activity of the E2F-Pocket protein complexes [62,63], adding another, yet unexplored level of complexity to the aforementioned connections between CDKs and HR genes. Taken together, a better apprehension of the mechanisms governing DNA repair gene transcriptional regulation of the different E2F family members could open novel therapeutic avenues in cancer.

#### 2.1.3. Other Transcription Factors

The transcription factor FOXM1 has been suggested to transactivate RAD51 binding directly to its promoter and mediating resistance to chemotherapy [64]. FOXM1 is a pro-oncogenic transcription factor that is overexpressed in glioblastoma multiforme (GBM) and correlated with increased RAD51 expression. FOXM1 inhibition has been proposed as a potential therapeutic strategy in GBM.

Other identified RAD51 transcription regulators comprise EGR1, as an activator, and p53 as a repressor [65,66]. Interestingly, while the normal function of phosphorylated p53 is to repress RAD51 expression, the p53 contact mutant R280K loses this ability and does not reduce DNA-damage-induced RAD51 foci, possibly explaining the resistance to chemotherapy of p53-mutated cancers.

### 2.2. Chromatin-Mediated Regulation of RAD51 Gene Expression

The members of the Bromodomains and Extraterminal (BET) family, comprising 4 genes BRD2, BRD3, BRD4 and BRDt (specific to testis and ovary), are important transcriptional and epigenetic regulators that play key roles in embryogenesis and cancer [67]. They harbor two evolutionary conserved tandem bromodomains (BD1, BD2) and an extraterminal domain (ET) [68]. The BD domains bind to acetyl-lysin residues of DNA-associated proteins, such as histones or transcription factors [69]. The affinity of BRD proteins to multi-acetylated histones is high in general, to the point that BRD4 is considered one of the most abundant proteins on transcriptionally active chromatin. Originally characterized as a key regulator of gene expression during the M to G1 transition [70], BRD4 was found to promote self-renewal of embryonic stem cells [71]. More recently, it was reported to be abundant at enhancer regions and particularly at super-enhancers (SE), which gather several enhancers of highly active genes, including HR genes such as BRCA1 and RAD51 [72,73].

The current model proposes that BRD4 stabilizes transcription factors at enhancers and activates transcription. Through its ET domain, BRD4 binds to various histone modifiers such as the demethylase JMJD6, the methyltransferase NSD3 and the SWIF/SNF enzyme. The recent discovery that super-enhancers in cancer cells are associated with major oncogenes, such as the MYC gene, has identified the BET protein family as a valuable drug target in cancer and fostered the development of several inhibitors (Table 2).

However, BRD4 is also an important actor of the mediator complex, which modulates RNA pol II activity, at least in part through the recruitment of the P-TEFb (CCNT1/CDK9) elongation factor. Hence, the inhibition of BRD4 could have effects similar to those of CDK9 inhibition [68,74,75]. Indeed, several studies proposed that the depletion or pharmacological inhibition of BRD4 directly impacted the expression of HR genes, such as RAD51, BRCA1 and BRIP1, and enhanced the sensitivity of cancer cells to PARP inhibitors [72,73]. These results were consolidated by chromatin immunoprecipitation (ChIP) results showing that BRD4-binding to the RAD51 promoter was indeed lost in breast cancer cell lines treated with the BRD4 inhibitor JQ1 and correlated with the loss of RAD51 mRNA expression [73]. These data were corroborated by a study on ovarian cancer cell line models showing that BET inhibitor INCB0544329 synergized with the PARPi Olaparib and resulted in reduced expression of BRCA1 [76]. Altogether, these data suggest that BET inhibitors and more specifically BRD4 inhibitors are interesting leads to induce BRCA-deficiency in cancer cells. Thus, results of ongoing clinical trials using BET-i should be followed with attention.

CHD4 (chromodomain helicase DNA-binding protein 4) is another modifier that has been proposed to impinge on the expression of HR genes. CHD4 is a highly conserved ATPase forming the core subunit of the nucleosome remodeling NuRD complex. NuRD acts both as a transcription repressor and transcription activator and has been shown to play a role in DNA repair [77]. Indeed, CHD4 was shown to be necessary for the recruitment of BRCA1 upon DNA damage in U2OS cells [78,79]. Furthermore, in glioblastoma cells, CHD4 was suggested to regulate RAD51 expression directly by binding to the RAD51 promoter and increase the acetylation of histone H3 at lysine 9 (H3K9Ac) [80], possibly through an interaction with the CBP and p300 co-activators. In line with this model, these acetyl-transferases have been shown to acetylate histones H3 and H4 at the RAD51 and BRCA1 promoters in H1299 human lung cancer cells [81], and CHD4 was shown to interact with p300 in a context of DNA damage [82]. However, it is not yet fully clear whether the CHD4/p300 interaction is critical for RAD51 upregulation or whether it may play a role in the resistance of cancer cells to DNA-damaging treatment.

## 3. Post-Translational Regulation of RAD51

Once RAD51 is transcribed and translated, its functions are orchestrated by post-translational modifications such as phosphorylation and ubiquitination on specific amino acids (Figure 4). A better understanding of these events could give leverage to modulate its activation or stabilization with a potential impact on anti-cancer treatment strategies.

### 3.1. Post-Translational Modifications Leading to Activation of RAD51

Phosphorylation by Polo-like kinase 1 (PLK1) and Casein Kinase 2 (CK2): In irradiated HeLa cells, RAD51 is first phosphorylated at Ser14 by PLK1, which primes for the phosphorylation at Thr13 by CK2 [83]. Since RAD51 binding with the FHA domain of NBS1 requires the phosphorylation at both S14 and T13, these sequential phosphorylations could represent important activating steps [83]. Indeed, the RAD51/NBS1 interactions are essential for RAD51 loading onto DSB sites in a BRCA2-independent manner, possibly promoting resistance to IR and PARPi in BRCA2-mutated HeLa cells. It is tempting to speculate that a combined therapy using inhibitors against both PLK1 and PARP or against CK2 and PARP may be an effective approach to improve the prognosis of BRCA-defective cancers.

However, it is unclear whether and through which signals PLK1 and CK2 are activated in response to DSB [84]. PLK1 overexpression has been associated with resistance to doxorubicin, paclitaxel, metformin and gemcitabine via several mechanisms, such as microtubule rearrangement, p53 inactivation or metabolic changes in cell culture and xenograft models [85]. Additionally, the Ser/Thr protein kinase CK2 is constitutively active and ubiquitously expressed, but its role in the DNA damage response is unclear. CK2 is primarily seen as an essential regulator of transcription, including that of major cancer transcription activators, such as MYC or β-catenin. A number of CK2 inhibitors have been described, some of which made it through preclinical stages. CX-4945 (Silmitasertib), a potent ATP competitor with an excellent selectivity, is one of the most promising. It is currently tested in phase I and II clinical trials in mono therapy or in combination with cisplatin and gemcitabine. The results of these combination trials could be highly revealing [86], and it should be interesting to investigate the role of RAD51 in the response.

Phosphorylation of RAD51 by CHK1: The CHK1 kinase has been proposed to contribute to the formation of RAD51 foci after genotoxic treatment. Attenuation of CHK1 expression by siRNA transfection was shown to strongly reduce RAD51 foci formation upon UV light irradiation, while CHK2 inactivation had no effect [87]. RAD51 foci formation depends on its phosphorylation at T307 and T309, which are consensus CHK1 phosphorylation sites. Accordingly, T307-309 phosphorylation and RAD51 foci formation were reduced in cells treated with a CHK1 inhibitor. The authors determined on human pancreatic cancer and osteosarcoma cell line models that CHK1 inhibition affected the de novo formation of RAD51 oligomers but not their stability [88,89]. Interestingly, the binding and recruitment of RAD51 by BRCA2 appeared to be dependent on its phosphorylation by CHK1 at T307, giving some mechanistic insight into the impact of CHK1 on RAD51 oligomerization [89]. Taken together, we suggest that the treatment of cancer cells with CHK1 inhibitor could not only affect the replication checkpoint, but could also interfere with RAD51 foci formation and, thus, induce HR deficiency. However, most preclinical and clinical trials have focused on the use of CHK1-i to potentiate replication poisons and induce catastrophic replication stress [90,91,92].

Deubiquitination by UCHL3: UCHL3 is a protease of the deubiquitinating enzyme family that catalyzes the removal of ubiquitin moiety from RAD51. RAD51 deubiquitination by UCHL3 was observed both in vitro and in cellulo and shown to be critical for BRCA2–RAD51 interaction, RAD51 recruitment to DSB sites, foci formation, as well as RAD51-mediated D-loop formation [93]. The key role played by UCHL3 in RAD51 activation is illustrated by the dramatic decrease in HR and PARPi sensitivity in UCHL3-depleted cells [93]. Hence, UCHL3 could be a valuable target to induce pharmacological HR deficiency.

### 3.2. Post-Translational Modifications Leading to RAD51 Inactivation

Ubiquitination of RAD51: Several factors control the ubiquitin-dependent RAD51 protein stability. These include two F-box domain protein and an RING-type E3 ligase, which appear to act at different time-points. The F-box DNA helicases FBH1 and FBX05 (also called EMI1) actively assemble SCF (SKP1-CUL1-F-box protein) complexes that act as E3 ubiquitin ligases and target RAD51 for degradation. FBH1 ubiquitinates RAD51 at K58 and K64, resulting in its accumulation in the cytoplasm and a reduction of HRR efficiency after DSB induction [94,95,96,97,98]. FBX05 (EMI1) was the top-hit of an siRNA screen targeting human F-box genes with the aim to identify mediators of olaparib (PARPi) resistance [99]. In this study, FBX05 was the only F-Box protein to co-immunoprecipitate with RAD51. Its silencing dramatically increased steady state levels of the RAD51 protein and induced the accumulation of RAD51 foci [99].

The RING-type E3 ligase RFWD3 binds to and ubiquitinates RPA2 in DNA-damage-induced foci. It was thus proposed to regulate levels of RPA2 bound to ssDNA [100,101] and to play a role in HR [99]. It was later shown that RFWD3 also promotes the ubiquitination of RAD51, reducing its binding to ssDNA and increasing its turnover in DNA damage foci [102]. Hence, whereas FHB1 and FBX05 seem to act as global regulators of RAD51 protein levels, RFWD3 appeared to be more specifically involved in the regulation of RAD51 levels bound to ssDNA during the DNA damage or DNA replication stress.

### 3.3. RAD51 Loading and Stability at DSB Site by Protein–Protein Interactions

As described earlier, RAD51 loading at DSB sites and the subsequent RAD51 homopolymeric-filament assembly, stabilization and timely disassembly are finely tuned steps that determine HR efficiency (Figure 5). A number of interactors and regulators have been identified and more might be discovered in the future.

RAD51 recruitment: In an HR-proficient context, RAD51 is actively recruited by a tri-protein complex formed by BRCA1, PALB2 and BRCA2. Of this triad, BRCA2 actively loads RAD51, while the two other partners stabilize the complex. This explains why RAD51 foci formation is reduced or absent in BRCA1 or PALB2 defective cells. However, alternative or safeguard mechanisms have been described to restore RAD51 recruitment in the absence of BRCA1. These range from the loss of expression of 53BP1, the phosphorylation of PALB2 by ATR and the binding of PALB2 by RNF168, which all favor PALB2 activation and accumulation onto DNA damage sites, where it promotes RAD51 recruitment by BRCA2 [103,104,105,106]. These observations clearly showed that the intervention of RNF168 could serve as an auxiliary backup mechanism in a BRCA-deficient context [107].

RAD51 paralogs: RAD51 is a highly conserved gene going back to recA in bacteria, which evolved during the early stage of eukaryotic evolution into the RAD51 and DMC1 (meiosis specific genes) subfamilies. In higher eukaryotes, the family comprises seven paralogs, RAD51, RAD51B, RAD51C, RAD51D, XRCC2, XRCC3, and the later identified SWSAP1 [108,109,110].

The RAD51 paralogs share only 20–30% identity at the protein level with RAD51, essentially in the ATP-binding Walker A and B domains and show comparable similarity to each other [109]. The biological functions of RAD51 paralogs have long been questioned. First, answers came from the yeast that possesses two paralogs, rad55 and rad57, which form heterodimers that interact with RAD51 and stimulate strand exchange reactions [111]. In higher eukaryotes, genetic studies on chicken cell lines showed that defective RAD51 paralogs were associated with increased sensitivity to DNA-damaging agents, suggesting a role in HR [111,112,113,114]. The importance of the RAD51 paralogs was further underlined by genetic studies showing an increased risk of developing breast and/or ovarian cancer in women bearing RAD51C, RAD51D or XRCC2 mutations [115,116,117]. This association was confirmed in breast and/or ovarian cancer families and cohorts with no known BRCA1 or BRCA2 mutations, leading to the proposition to include RAD51D and RAD51C to the list of genes screened for breast cancer predisposition [118,119,120].

In 2001, Masson and coworkers showed that the five human paralogs assembled in two distinct complexes: the first one, BCDX2, is formed of RAD51B, RAD51C, RAD51D and XRCC2, while the second one, identified as CX3, was comprised of RAD51C and XRCC3 [121,122,123]. More recently, SWSAP1, number six of the gang and non-classical RAD51 paralog, was shown to form a heterodimer with SWS1 termed the Shu complex [124,125,126]. Whereas the RAD51 paralogs are known to promote RAD51-mediated activities, their precise mechanism of action is not yet completely clear. To decipher their respective roles, several groups have performed conditional knock-out studies of the different genes. Using drug sensitivity and RAD51 foci formation as read-outs, they observed phenotypes of variable severity. The strongest impact was observed for RAD51C and RAD51D disruption, whereas RAD51B knock-out only moderately affected HR [127]. The increased impact of RAD51C and RAD51D was explained by the disruption of the CX3 and BCDX2 complexes and associated destabilization of the XRCC3 and XRCC2 proteins. It was also demonstrated that the CX3 and BCDX2 complexes operate at different stages of HR. Both act downstream of BRCA1 and BRCA2 recruitment to damage sites, but BCDX2 acts upstream of RAD51 recruitment, while CX3 has been suggested to intervene after RAD51 homopolymer formation and possibly stabilize them [123,128]. However, results showing that the CX3 complex binds directly to ssDNA suggest that it could also act prior to RAD51 foci formation [122].

SWSAP1: In yeast and worms, disruption of the Shu complex resulted in increased mutagenicity and impaired HR [124,129,130]. Unlike other RAD51 mediators, disruption of the Shu genes primarily impacted on MMS sensitivity and not on other DNA-damaging agents such as IR, hydroxyurea or the Topoisomerase 2 inhibitor etoposide [131]. Consistent with a conserved function of the Shu complex in humans, siRNA depletion of SWS1 and SWSAP1 in human cells resulted in increased MMS sensitivity and reduced RAD51 foci formation [124,125].

PARI: PARI is an UvrD-like helicase-domain containing protein that is recruited on DNA by SUMO-modified PCNA and shown to actively reduce RAD51 nucleofilament formation [132]. Indeed, the knock-down of PARI significantly increased RAD51 foci formation in both basal and DNA-damaging conditions [132]. Hence, PARI has been proposed to act as a negative regulator of HR by disassembling toxic RAD51 nucleofilaments. It is, however, not known whether the disassembly of RAD51 filaments by PARI is associated with a post-translational modification of RAD51.

BLM: the BLM gene codes for a DNA helicase of the RecQ family, which has been shown to interact with RAD51. RAD51 foci formation appeared constitutively activated in BLM-mutated cells and was suppressed by the ectopic expression of functional BLM, suggesting that BLM negatively regulates RAD51 foci formation [133].

## 4. Pharmacological Inhibition of RAD51

The important role of RAD51 in DNA repair made it an attractive pharmacological target in cancer treatment and has motivated investigators to search for small molecule inhibitors that could disrupt its functions. At least seven compounds antagonizing RAD51 efficiently have been reported.

B02 has been discovered as part of a high-throughput in vitro screening assay scoring for DNA strand exchange. It specifically inhibits human RAD51, but not the E. coli RecA protein [134]. It disrupted RAD51 binding to DNA, inhibits nucleoprotein filament formation and greatly increases the sensitivity of cancer cells to a wide range of genotoxic drugs, including the DNA cross-linking agents cisplatin and mitomycin C, as well as Topoisomerase 1 (Topotecan) and Topoisomerase 2 (doxorubicin) inhibitors. Further studies have shown interesting effects in combination with MAPK inhibitors on melanoma cells or with proton therapy [135,136]. The therapeutic potential of B02 was confirmed in vivo, where it substantially increased the anticancer potential of cisplatin and notably showed little toxic side effects. Indeed, treated mice showed limited weight loss and no macroscopic modifications of the kidney or liver [136]. These results encouraged the development of a new chemical analogue of B02, B17, which presents greater efficacy than B02 in vitro [137]. However, in vivo evaluation has not been reported yet.

Compounds RI-1 and R7a were isolated on the basis of their capacity to inhibit RAD51 binding to ssDNA [138]. Tested on cell cultures, RI-1 showed potent anti-proliferation activity; however, its short half-life appeared as a strong limit to its in vivo efficacy due to low bio-availability [139].

Compounds IBR2 and derivative IBR120 were isolated on the basis of their capacity to interfere with RAD51 oligomerization and to potentiate its degradation by the proteasome. IBR2 and IBR120 were shown to efficiently inhibit cell growth in a broad set of cancer cell lines [140,141]. In vivo, IBR2 has been tested in single treatment on an imatinib-resistant murine chronic myelogenous leukemia (CML) model, where it significantly prolonged animal survival, without any notable signs of toxicity [140].

Last but not least, the development of compound CYT-0851 is of interest. Firstly, because it is an oral inhibitor that has shown anti-cancer activity on lymphoma as well as the 4T1 mammary tumor in preclinical mouse models [142]. Secondly, because it is, to our knowledge, the only RAD51-i tested in a clinical trial (Appendix A).

Altogether, these results suggest that there could be a bright future for small molecule inhibitors of RAD51. However, further tests will be needed to determine in which genetic contexts or tumor types they will prove most effective. Moreover, similarly to other targeted therapies, it is now necessary to identify the best synergistic drug combinations to use with these RAD51 inhibitors. Lastly, induced toxicity and mutagen effects will be critical parameters to take into account for these potent inhibitors.

## 5. Discussion

Due to its prominent role in DNA double-strand break repair, as well as in the protection and repair of stalled replication forks, RAD51 is a crucial actor in the maintenance of genomic stability. Its expression and activity must be tightly controlled in normal cells. Indeed, insufficient levels of RAD51 could cause an accumulation of unrepaired DNA breaks. Conversely, in excess, RAD51 could provoke hyper-recombination and genetic instability. The question about the consequences and advantages of the strong overexpression of RAD51 in a large fraction of tumors at both the RNA and protein levels remains puzzling. As genomic instability is almost inherent in cancer cells, it is tempting to speculate that over-activation of RAD51 could be part of a compensation mechanism to protect cells from excessive genetic instability. Remarkably, RAD51 is strictly essential for cancer cell survival, as illustrated by the 98% cell lines showing elevated dependency scores for RAD51 gene knock-out (the DepMap database www.depmap.org, accessed on 15 March 2021). In comparison, BRCA1 and BRCA2, two important partners of RAD51, are strictly required for the survival of only 84.5 and 84.7% cell lines, respectively (CERES scores below −0.5). It is of note that BRCA1-mutated cell lines are among those showing the highest dependency scores to RAD51. In fact, a number of BRCA1 or BRCA2-deficient cell models do not show complete obliteration of RAD51 foci when exposed to a genotoxic insult, but rather an attenuated response. This can be explained by maintained or restored recruitment of RAD51 onto DNA lesions even in the absence of BRCA1 or BRCA2 due to compensatory mechanisms which are not yet fully understood. Thus, RAD51 is a crucial DNA repair actor in a large majority of cancers, notwithstanding their BRCA proficiency or deficiency. Consequently, it has been shown to be an interesting biomarker, as *RAD51* expression levels have been proposed to be potentially predictive of resistance to treatment [143,144]. Additionally, assays aimed at determining the capacity of tumor cells to form RAD51 foci when challenged with a DNA-damaging treatment have been proposed as biomarkers of response to PARP-i (NCT04780945) [145]. Furthermore, RAD51 appears as a valuable therapeutic target, especially in tumors prone to accumulate DNA breaks, such as cancers showing strongly rearranged genomes. The design of pharmacological approaches that would efficiently impair or attenuate RAD51 recruitment onto DNA lesions and/or the formation of presynaptic filaments is, in our opinion, the therapeutic strategy of choice, especially in combination with DNA-break-inducing agents. As described above, approaches are aiming at downregulating the mRNA expression of genes belonging to the HR pathway, among which *RAD51* is being actively studied. Given the central role of RAD51 in HR, inhibitors that directly target the protein are, as discussed above, interesting leads to be followed (Appendix A) [146,147]. Moreover, the future may also bring about further approaches based on inhibitors targeting either enzymes that activate RAD51 or target its cofactors and interfere with homopolymeric filament formation or stabilization. Which approach will be most effective remains undetermined, but we are convinced that targeting RAD51 directly or its cofactors will have great clinical potential in the future.

## Figures and Tables

**Figure 1 cancers-13-02930-f001:**
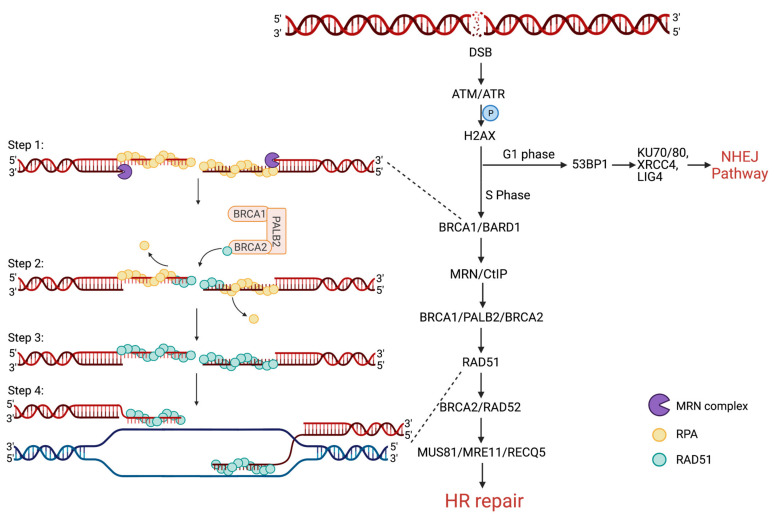
Representation of DNA double-strand break repair choice in the S/G2 phase between homologous recombination non-homologous end joining. The scheme details the steps leading homologous strand invasion and D loop formation. Step 1: 3′ end resection by the MRN complex, RPA loading to protect single-strand DNA (ssDNA) from degradation. Step 2: BRCA2 is recruited by BRCA1/PALB2 and loads RAD51 onto the ssDNA overhangs and RPA is dislodged. Step 3: formation of the RAD51 Homopolymeric filament which wind around the ssDNA. Step 4: invasion of the homologous chromatid and search for homology in view of recombination and repair.

**Figure 2 cancers-13-02930-f002:**
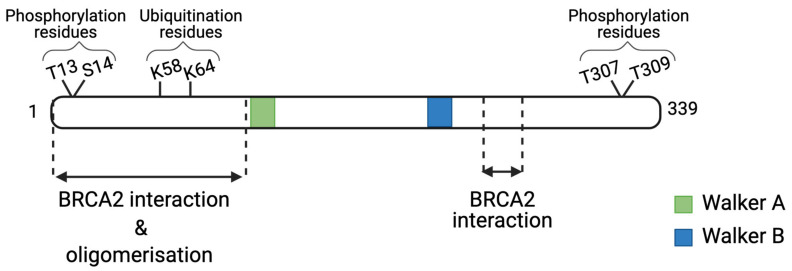
The schematic view of the RAD51 protein structure. The Walker A and B domains of the RAD51 family are depicted as green and blue boxes. The different interaction and post-translational modification sites are represented.

**Figure 3 cancers-13-02930-f003:**
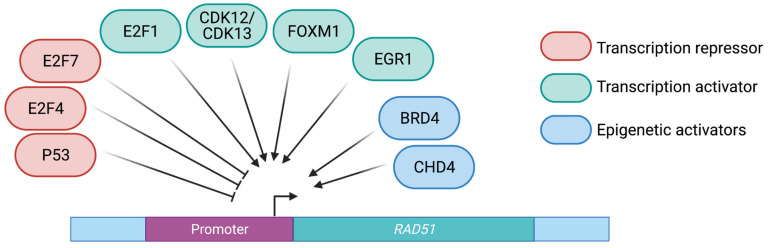
Identified regulators of RAD51 transcription. The DNA sequences containing the RAD51 promoter and the gene are represented as a purple box and a green box, respectively.

**Figure 4 cancers-13-02930-f004:**
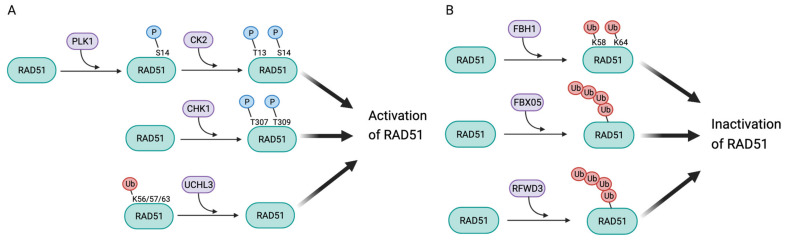
Post-translational modifications that impinge on RAD51 function. (**A**) Modifications leading to the activation and loading of RAD51: phosphorylation cascade at Ser14 and Thr13 by PLK1 and CK2, phosphorylation at Thr307/309 by CHK1, ubiquitination at Lys56/57/63 by UCHL3. (**B**) Modifications leading to the inactivation of RAD51: ubiquitination at Lys58/64 by FBH1, polyubiquitination by FBK05 or RFWD3 leading to the degradation of RAD51 by the proteasome. Blue bubbles (P) represent phosphorylation, red bubbles (Ub) represent ubiquitination.

**Figure 5 cancers-13-02930-f005:**
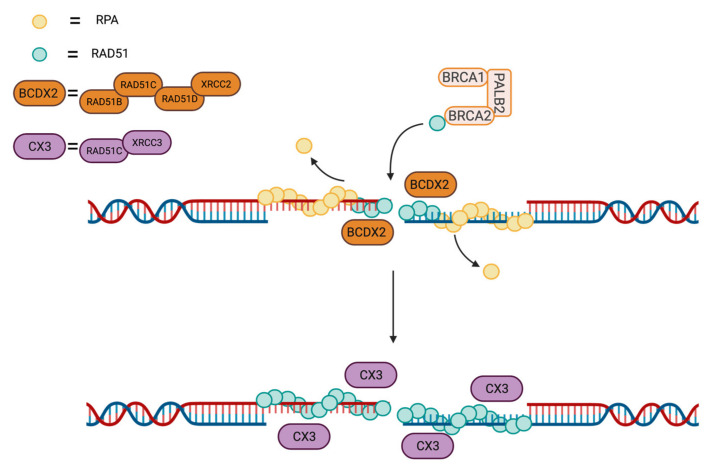
Intervention of RAD51 cofactors in RAD51 loading and stabilization of the filaments. The BCDX2 intervenes at early stages of RAD51 loading, while CX3 stabilizes the filaments onto the ssDNA.

**Table 1 cancers-13-02930-t001:** Regulators of RAD51 at the transcriptional and post-translational levels. Abbreviation: RNA polymerase II (RNA pol II).

Regulator of RAD51	Function	Level of regulation	Effect on RAD51	Drug Target
CDK12/CDK13	RNA pol II regulator	mRNA expression	Activator	Yes
E2F1	Transcription Factor	mRNA expression	Activator	No
E2F4	Transcription Factor	mRNA expression	Repressor	No
E2F7	Transcription Factor	mRNA expression	Repressor	No
FOXM1	Transcription Factor	mRNA expression	Activator	No
EGR1	Transcription Factor	mRNA expression	Activator	No
P53	Transcription Factor	mRNA expression	Repressor	No
BRD4	Transcription Activator	mRNA expression	Activator	Yes
CHD4	Nucleosome Remodeler	mRNA expression	Activator	No
PLK1	Kinase	Post-translational	Activator	Yes
CK2	Kinase	Post-translational	Activator	Yes
CHK1	Kinase	Post-translational	Activator	Yes
UCHL3	Deubiquitinatinase	Post-translational	Activator	No
FBH1	Ubiquitin ligase	Post-translational	Repressor	No
FBX05	Ubiquitin ligase	Post-translational	Repressor	No
RFWD3	Ubiquitin ligase	Post-translational	Repressor	No
RAD51C	RAD51 Cofactor	Homofilament Loading/stabilisation	Activator	No
RAD51B	RAD51 Cofactor	Homofilament Loading	Activator	No
RAD51D	RAD51 Cofactor	Homofilament Loading	Activator	No
XRCC1	RAD51 Cofactor	Homofilament stabilisation	Activator	No
XRCC2	RAD51 Cofactor	Homofilament Loading	Activator	No
SWSAP1	RAD51 Cofactor	Homofilament stabilisation	Activator	No
PARI	DNA Helicase	Homofilament stabilisation	Repressor	No
BLM	DNA Helicase	Homofilament stabilisation	Repressor	No

**Table 2 cancers-13-02930-t002:** Molecules targeting potential regulators of RAD51 expression and function tested in clinical trials.

Drug	Target	Trial phase	Trial Status	Results	Trial Number
CYT-0851	RAD51	Phase 1/2	On going	Not reported	NCT03997968
Dinaciclib	Inhibition of multiple CDKs	Phase 1	On going	Tolerable in combination with veliparib	NCT01434316
				Anti-tumor activity is limited in non-BRCA carriers	
INCB054329	Inhibition of BET proteins	Phase 1/2	Completed	30% Stable disease	NCT02431260
				43% Progressive disease	
SYHA1801	BRD4	Phase 1	On going	Not reported	NCT04309968
PLX51107	BRD4	Phase 1	Completed	Tolerable except for patients with extensive hepatic metastasis	NCT04022785
		Phase 1	On going	Not reported	NCT04022785
BMS-986158	Inhibition of BET proteins	Phase 1	On going	Not reported	NCT03936465
AZD5153	BRD4	Phase 1	Completed	Tolerable	NCT03205176
GSK525762	Inhibition of BET proteins	Phase 1	Completed	Manageable tolerability	NCT01587703
Volasertib	PLK1	Phase 3	Completed	Did not meet primary objectif in combination with cytarabine	NCT01721876
Onvansertib	PLK1	Phase 2	On going	Not reported	NCT03414034
TAK-960	PLK1	Phase 1	Completed	Terminated due to lack of efficacy	NCT01179399
CYC140	PLK1	Phase 1	Completed	Not reported	NCT03884829
NMS-1286937	PLK1	Phase 1	Completed	Tolerable	NCT01014429
CX-4945	CK2	Phase 1/2	Completed	Not reported	NCT02128282
		Phase 1/2	On going	Not reported	NCT03904862
MK-8776	CHK1	Phase 2	Completed	Similar result as control arm	NCT01870596
SRA737	CHK1	Phase 1/2	Completed	Tolerable	NCT02797964
		Phase 1/2	Completed	Tolerable	NCT02797977
LY2603618	CHK1	Phase 1/2	Completed	Toxicy when combined with pemetrexed+Cisplatin	NCT01139775
Prexasertib	CHK1	Phase 2	On going	Tolerable, only modest activity in BRCA mutant HGSOC patients	NCT02203513
		Phase 2	On going	Not reported	NCT03414047

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
