# Peer review of "Regulation of RAD51 at the Transcriptional and Functional Levels: What Prospects for Cancer Therapy?"

_cancers, 2021, doi:10.3390/cancers13122930_

Round 1

Reviewer 1 Report

In this very comprehensive review, Orhan et al gathered information concerning RAD51 regulation and its role in DNA repair mechanisms. They assessed the potential value of RAD51 as a therapeutic target to treat cancer cells.

Starting with the well know DNA repair mechanisms of double strand breaks the authors thoroughly describe the role of RAD51 in the HR repair. Further, they report the critical role of RAD51 in replication fork reversal.  The downregulation of RAD51 is considered a possible tool to overcome treatment resistance of tumors towards DNA-damaging agents. The regulation on of RAD51 by numerous transcription factors as well as the chromatin-mediated RAD51 gene expression are described followed by post-translational modifications leading to either activation or inactivation of RAD51. The manuscript ends in a chapter about pharmacological RAD51 inhibition and the authors discuss possible mechanisms that might play a role in the development of targeted therapies such as small molecules in various cancer types and also consider possible combination therapies.

The manuscript is well written and the reader can follow although a lot of information is delivered and numerous abbreviations are used- which is understandable and crucial considering the long names of genes, proteins etc.

However, some passages are not as “fluid” and might be improved.

Some minor mistakes should be revised to enhance the quality of the manuscript and to make it easeier to understand:

DSB in title: I would recommend to avoid abbreviations in the manuscript heading

Fig. 1: Mark 3‘ and 5‘ ends in the scheme. Figure caption (Use abbreviations in brackets)

Use colons when introducing the relevant step:

Step1:  …. Step2:….

Line 102 – 104 : Long sentence.

Passage 106 – 119 double

Figure 3: Figure caption is very short. Explain in a little more detail. Regulator identified by who? Parhaps add a strand structure to indicate that the box is DNA.

135 funny sentence

150 a particularly…

154: siRNA abbreviation not introduced yet

160: choice of words makes it difficult to follow long sentences with lots of abbreviations. Use more verbs such as …compared to…and what lists are meant by the author?

170:  Try to use less “of” would make the manuscript easier to read.

2.1 Why is this passage devided in so many paragraphes? Perhaps more sub-headings might help.

209 check tense, e.g. past tense vs. present

220 proteins (plural)

222 stem cells (plural)

229 the BET protein family

233 Similar effects or effects similar to

238 Here it would be interesting to mention  what kind of cancer cells were used in the studies cited? Cell lines? Primary cells? What cancer types?

240 What is meant by BRCAness?

Figure 4: A) Cannot find CK1.  Typing error? CK2? Add legend showing blue bubbles (P) and red bubbles (UB) similar to Figure 3.

269 Abbreviation IR not introduced yet

274: Is this the authors opinion strengthened by the reference (84) or cited from the literature?

276: Was this shown in cell culture or clinical samples?

277: “On the other hand….”This phrase should be used when there is a “On the one hand…” initially.

Passage starts about PLK1 and CK2, followed by a new paragraph in line 273. Another passage about CHK follows in line  287. Perhaps more sub-headings would help to structure the manuscript.

290 - 294: What was the experimental set up concerning the cancer cell origin?

297: Does this statement refer to treatment of cancer cells in patients or in cell culture? Please explain the hypothesis in more detail. Is this conclusion drawn by the author’s themselves?

306: Here, and also at other passages throughout the manuscript, it does not become clear whether the authors conclude the statement from the literature cited above or whether they want to postulate the hypothesis.

317: in which study exactly?

323: …”its binding” double

385: homologous recombination =HR. It certainly helps to use the entire phrase instead of abbreviation only throughout the manuscript. However, the use should be consistent. See also lines 72 and 79.

430: However, comma missing

Author Response

Thank you.

Reviewer 2 Report

This a very well oriented review. The data is well presented and it si easy to read. 

paragraph from line 92 to 105 is repeated in lines 106-119. One of these paragraphs should be deleted.

Author Response

Thank you.

Reviewer 3 Report

The authors have written a thorough review about RAD51 function and potential ways of interfering with RAD51. They summarize published compounds/small molecules that may impact RAD51 and thus affect cancer cells. The article is accompanied by some clear, describing figures, that help the reader to better understand the text.

My first concern regards the title; apart from the abbreviation DSB, which should preferably be avoided in the title and instead written as “double strand breaks”, I think the title is misleading. The review mainly concerns RAD51 regulation and function, and the potential prospects for cancer therapy only constitutes a minor part of the article. Either the title should be changed or, preferably, the discussion about potential clinical use of targeting RAD51 should  be expanded.

My second concern regards the level of details in the review. The text is comprehensive and educational, but the details make it hard to keep up with. The included figures are very helpful, but in general the text would benefit from more figures or maybe summarizing tables or flow charts. That would help the reader to understand the complex regulation of RAD51 and its effects.

Furthermore, I am totally missing the discussion about RAD51 as a potential biomarker for HR capacity (line 128). It is briefly mentioned previously (line 119), but does not show up later in the text. This would really be of great clinical interest.

Lastly, section 5 “conclusion” is more of a discussion than a conclusion. This section could preferably be expanded and include a discussion of potential clinical use of RAD51 inhibitors. Eg., could a RAD51i potentially help overcome PARPi resistance? In what other ways would targeting RAD51 be of clinical utility? This discussion is nicely started in this article, but is very brief.  

Detailed comments follow below:

  1. Line 36: explanation of abbreviation is missing
  2. Lines 106-119 are a copy of lines 92-105
  3. Line 128: RAD51 as a clinical biomarker for HR capacity – where is this discussion?
  4. The first text part in subsection 2.1 is very detailed, but is somehow separate from the following parts. It would benefit from being integrated better with the rest of the text.
  5. Subsections 2.1, 2.2, and 3.3 would all benefit from more explanatory figures or flow charts to make the text easier to follow
  6. It would be great if sections 2 and 3 were ended with a small summary as in section 3

Author Response

Thank you.

Reviewer 4 Report

In general, the review article seems suitable for publication. I only have minor comments:

  1. There is a duplicated paragraph in page 4
  2. In section transcriptional control of RAD51: there is a long description of CDH7/CDK9, CDK12/CDK13 (lines 133-157) in which the connection of these kinases with RAD51 is absent. I would have preferred that the authors first described documented evidence on the role of these regulators on RAD51 expression and thereafter mention their mechanisms of action and their impact on HR.

A role for E2F1 in RAD51 regulation is mentioned (lines 178-184) but this has not been included in Figure 3.

  1. Chromatin-mediated Regulation of RAD51 gene expression: alike mentioned above, lines 213-233 describes mechanism of action of BRD4 and only at the end its connection is described with RAD51.
  2. Another point I think could improve the manuscript is to add more specific mentions of on-going clinical trials given that is repeatedly mention the value of inhibitors of RAD51 and related regulators as anti-cancer drugs. For example: the development of drugs is still in preclinical analysis? What are the most promising targeted therapies, if any? Etc..

Author Response

Thank you.

Round 2

Reviewer 3 Report

According to my opinion, the authors have responded to and made the suggested changes in the manuscript. I therefore support that the manuscript is accepted and published in its revised form (version 2).